# HH-Codec: High Compression High-fidelity Discrete Neural Codec for Spoken Language Modeling

**Rongkun Xue** [1]  **Yazhe Niu** [2 3]
**Shuai Hu** [4]  **Zixin Yin** [4]  **Yongqiang Yao** [5]  **Jing Yang** [1]

## Abstract

Discrete speech tokenization is a fundamental component in speech codecs. However, in large-scale speech-to-speech systems, the complexity of parallel streams from multiple quantizers and the computational cost of high-time-dimensional codecs pose significant challenges. In this paper, we introduce *HH-Codec*, a neural codec that achieves extreme compression at 24 tokens per second for 24 kHz audio while relying on single-quantizer inference. Our approach involves a carefully designed Vector Quantization space for Spoken Language Modeling, optimizing compression efficiency while minimizing information loss. Building on this, we propose an asymmetric encoder-decoder architecture (Audio-VQ-Mel-Audio) that leverages dual supervision and progressive training to enhance reconstruction stability and fidelity. *HH-Codec* achieves state-of-the-art performance in speech reconstruction with an ultra-low bandwidth of 0.3 kbps. We further evaluate its effectiveness in codebook utilization and generative model adaptation, with extensive ablations validating the necessity of each module. *HH-Codec* is available at `https://github.com/opendilab/HH-Codec`.

## 1. Introduction

Unlike traditional waveform and parametric codecs like Opus (Valin et al., 2012) and Enhanced Voice Services (Dietz et al., 2015), neural audio codecs have gained attention for applications in many areas like speech emotion analysis (Felix et al., 2021), accent conversion (Nguyen et al., 2023), and speech-to-speech translation (Popuri et al., 2022). Recently, the rise of Large Language Model (LLM)-based audio generation (Achiam et al., 2023), latent diffusion-based Text-to-Speech (TTS) (Du et al., 2024), and multimodal speech integration (Défossez et al., 2024) has further underscored the need for efficient discrete acoustic representations (Défossez et al., 2024; Hsu et al., 2021). These methods transform audio into token representations compatible with LLMs, typically categorized as Automatic Speech Recognition (ASR)-based semantic tokens or Vector Quantization (VQ) -based acoustic tokens.

Specifically, the former approach maps audio to semantic features through methods such as HuBERT (Hsu et al., 2021), which employs the BERT-based self-supervision; Wav2Vec2 (Baevski et al., 2020), a method for pretraining ASR models on unlabeled data and fine-tuning with limited supervision; and CosyVoice (Du et al., 2024), which employs a supervised ASR model for semantic tokenization. While these methods achieve efficient compression, they often sacrifice acoustic details, particularly affecting emotional expressiveness captured by large models. Additionally, their scalability remains challenging, limiting practical applications. For the latter, originating from HiFi-GAN (Kong et al., 2020), researchers have developed discrete neural codecs that train encoder-decoder networks with reconstruction losses and VQ techniques (Défossez et al., 2022) to discretize hidden representations into compact code vectors. However, recent studies (Défossez et al., 2024; Ji et al., 2024; Li et al., 2024) point out a key challenge of VQ-based approaches: temporal compression ratios and quantizer counts are crucial for LLM-based audio tasks.

In contrast to natural language tokenizers like BPE (Sennrich, 2015) used in LLMs, which process approximately 100 tokens per 75 words (e.g., in GPT-4o (Achiam et al., 2023)), existing discrete neural codecs exhibit substantially higher token rates - DAC (Kumar et al., 2024) operates at 900 tokens/second while SpeechTokenizer (Zhang et al., 2024) requires 300. This significant disparity stems primarily from the use of multiple discrete VQ streams; for

---

[1]The School of Automation Science and Engineering, Xi'an Jiaotong University, Xi'an, China [2]Shanghai Artificial Intelligence Laboratory, Shanghai, China [3]The Chinese University of Hong Kong, Hong Kong SAR, China [4]SenseTime Research, Shanghai, China [5]Shanghai Jiao Tong University, Shanghai, China. Correspondence to: Jing Yang <jasmine1976@xjtu.edu.cn>, Yazhe Niu <niuyazhe314@outlook.com>, Rongkun Xue <xuerongkun@stu.xjtu.edu.cn>.

*Non-archival presentation at ICML 2025 Tokenization Workshop (TokShop)*, Vancouver, Canada. 2025.

instance, DAC employs 9 Residual Vector Quantization (RVQ) layers (Défossez et al., 2022), while SpeechTokenizer utilizes 4. To address the resulting model complexity, various solutions have been proposed, including VALL-E's non-autoregressive approach (Wang et al., 2023) and MusicGen's token interleaving technique (Copet et al., 2024), both aiming to optimize the processing of multiple VQ streams. However, these approaches still face considerable challenges in terms of computational efficiency and model complexity. Thus, the adoption of a single quantizer emerges as a promising alternative. This approach enables more consistent and seamless integration between speech and text models, and it significantly streamlines both training and inference processes.

However, previous single quantizers require relatively high token rates to maintain high-quality audio modeling. WavTokenizer (Ji et al., 2024) utilizes K-means clustering and random awakening strategies within the VQ space, achieving 75 tokens per second. This remains far less efficient than natural language tokenization. Similarly, Single-Codec (Li et al., 2024) uses a specialized BiLSTM architecture to preserve performance with a single quantizer, albeit only reconstructing the mel-spectrogram. The pursuit of extreme compression with single quantizers reveals two fundamental challenges. First, a constrained vocabulary struggles to capture rich semantic content in the codebook space, making the entire training especially difficult. Second, under low bandwidth, the audio reconstruction loss is even higher during the early stages of training and remains difficult to reduce; in this setting, applying adversarial training (Kong et al., 2020) to an under-trained codebook not only makes training unstable but also markedly degrades the final reconstruction quality. Our empirical analysis of tuning the training hyperparameters of WavTokenizer at ultra-high compression ratios yields several key observations, which more intuitively demonstrates these challenges:

① **Adversarial Training Collapse**: Reducing token counts in existing methods increases early-stage failure rates. When the bandwidth drops below 0.3 kbps, introducing additional scheduling strategies and lowering the learning rate becomes necessary to maintain stable training.

② **Limited Benefits from Larger Dataset**: Expanding from LibriTTS (Zen et al., 2019) train-clean-100 to train-clean-360 improves UTMOS (Saeki et al., 2022) by only 0.6 with 20 tokens but by 1.3 with 75 tokens.

③ **Low Codebook Utilization**: At an extreme compression rate of 0.3 kbps, an 8192-entry codebook achieves only 43 %, whereas at 0.75 kbps utilization exceeds 90 %.

④ **Severe Quality Degradation**: When the token rate drops below 30 per second, UTMOS decreases by 63%.

To overcome these limitations, we introduce *HH-Codec*, a discrete neural codec that achieves an unprecedented compression rate of 24 tokens per second while operating at an ultra-low bandwidth of only 0.3k bits per second (0.3kbps). Our solution incorporates three key innovations: Firstly, we propose SLM-VQ, a specialized VQ space designed for spoken language modeling, as illustrated in the middle part of Figure 1. This space aims to preserve critical semantic information and essential acoustic characteristics (e.g., emotion) while discarding redundant details to achieve high compression ratios. In doing so, we align the granularity of audio tokens with that of text tokens, enabling seamless integration with language models. Inspired by Zhu et al. (2024), SLM-VQ employs a frozen codebook with a learnable MLP to implicitly generate codes. And we replace the traditional straight-through estimator with a rotational trick (Fifty et al., 2024) for gradient backward. Additionally, the first-layer output of SLM-VQ is projected via a linear layer and optimized through HuBERT-based semantic distillation, ensuring effective feature extraction. More importantly, we employ multi-layer residual connections during training to enhance gradient flow and network optimization, while maintaining inference efficiency by utilizing only the first VQ layer - this design choice preserves both high compression ratios and model simplicity. That is to say, the second VQ layer serves as a "Virtual Class" (Chen et al., 2018; Darcet et al., 2024), serving to regularize and improve the compactness of the first layer.

Secondly, we adopt an asymmetric encoder–decoder architecture augmented with two critical enhancements: (1) a dedicated attention mechanism for capturing long-range audio dependencies, and (2) a dual-supervision scheme that simultaneously operates in both Mel-spectrogram and audio domains. This design leverages a more powerful decoder and incorporates dual supervised signals operating within both domains. To optimize training efficiency, we initialize the decoder with a pre-trained BigVGAN and freeze its weights during the initial training phase. Once the other parts of the network converge to a compatible state, we unfreeze and fine-tune the entire architecture to further refine its performance. Extensive experiments demonstrate that HH-Codec performs comparably to several state-of-the-art speech tokenizers across diverse datasets with the lowest tokens per second. Comprehensive ablation studies further validate the necessity of each component in our design. and validate its potential for spoken language modeling. Our contributions can be summarized as follows:

• We introduce a discrete neural codec that achieves excellent speech reconstruction with the lowest cost—24 tokens for 24 kHz audio at a just 0.3 kbps bandwidth.

• We design a speech-specific SLM-VQ that incorporates an asymmetric architecture, significantly enhancing stability

and performance under high compression conditions.

- Experiments show that *HH-Codec* performs on par with several leading speech tokenizers across various datasets. Ablations further confirm the necessity of each component and validate its potential for spoken language modeling.

## 2. Related Work

Neural acoustic codecs (Du et al., 2024; Kong et al., 2020; Siuzdak, 2023), as an indispensable component of large-scale speech models, are receiving widespread attention from the community. Current contributions can be broadly summarized into three categories: architecture, neural-network design, and codebook construction.

**Architectural Innovations** SpeechTokenizer (Zhang et al., 2024) integrates residual connections and knowledge distillation to jointly model acoustic and semantic information. In contrast, Moshi (Défossez et al., 2024) employs parallel multi-vector quantization (multi-VQ) to improve reconstruction fidelity, demonstrating that diverse quantizers can capture complementary signal characteristics.

**Encoder–Decoder Designs** Most recent methods use a wave-to-wave fully convolutional encoder—drawing on SEANet (Tagliasacchi et al., 2020) and SoundStream (Zeghidour et al., 2021). Decoder designs vary—from progressive upsampling schemes to attention-enhanced modules—but there is broad consensus that encoder capacity chiefly determines system performance. For instance, VOCOS (Siuzdak, 2023) predicts Fourier spectral coefficients to better leverage time–frequency inductive biases and align with human auditory perception, whereas BigVGAN (Lee et al., 2022) integrates periodic nonlinearities and anti-aliasing filters to embed inductive bias directly into waveform synthesis.

**Codebook Engineering** Borrowing ideas from image compression, modern speech codebooks aim to maximize representational capacity and utilization. WavTokenizer expands the VQ space and introduces tailored initialization schemes to prevent cluster collapse and uneven code usage. FSQ (Mentzer et al., 2023) takes a complementary route: it enforces code vectors to lie on a uniform hypercube grid, obviating commitment losses and exponential moving average (EMA) updates, and fully mitigating codebook collapse. SimVQ (Zhu et al., 2024) a novel method which reparameterizes the code vectors through a linear transformation layer based on a learnable latent basis.

## 3. Method

As illustrated in Figure 1, Our High Compression High-fidelity Codec (*HH-Codec*) builds on the VQ-GANs (Ji et al., 2024) framework. We integrate an advanced encoder for high-quality compression, an SLM-VQ for spoken language modeling, and a BigVGAN-based decoder for high-fidelity audio reconstruction. The complete architecture is optimized through a dual-supervised progressive training strategy. In the following sections, we will describe each component and training strategies in detail.

### 3.1. Notation

Throughout this paper, we use the following math notation.

- $z$ – input waveform sampled at 24 kHz;

- mel – Mel-spectrogram of $z$;

- $e$ – latent code produced by the encoder;

- $\hat{e}$ – quantized code reconstructed by SLM-VQ;

- $\text{mel}_{\text{rec}}$ – Mel-spectrogram reconstructed by the decoder;

- $\hat{z}$ – waveform reconstructed by BigVGAN;

- $\hat{\text{mel}}_{\text{rec}}$ – Mel-spectrogram of $\hat{z}$.

### 3.2. Encoder

Our encoder follows a convolutional architecture inspired by prior works (Défossez et al.; Ji et al., 2024). The initial layer consists of a 1D Convolution with a kernel size of 7, followed by four Conv. blocks. Each block incorporates a dilated convolution and a downsampling layer with skip connections and strided Conv. The channel count doubles after each downsampling operation. Following Zhang et al. (2024), we enhance semantic modeling by replacing conventional LSTM layers with BiLSTM after the Conv. blocks. A final 1D convolution with a kernel size of 7 projects the output into a 512-dimensional embedding space. To achieve aggressive compression, we employ a stride configuration of (8, 8, 4, 4) - significantly larger than previous approaches - enabling the encoding of 24 kHz raw audio into a compact sequence of 24 tokens per second. To improve training stability and facilitate Mel-spectrogram domain learning, we diverge from conventional acoustic codecs practices that rely on randomly sampled one-second clips. Instead, we employ longer training windows, which enhance generalization by preserving word-level contextual coherence.

### 3.3. SLM-VQ

To ensure sufficient modeling capabilities and prevent codebook collapse, we first adopt SimVQ (Zhu et al., 2024) as the foundational vector quantization (VQ) framework. Next, we introduce a multiple VQ structure that uses multiple residual layers during training but only a single layer

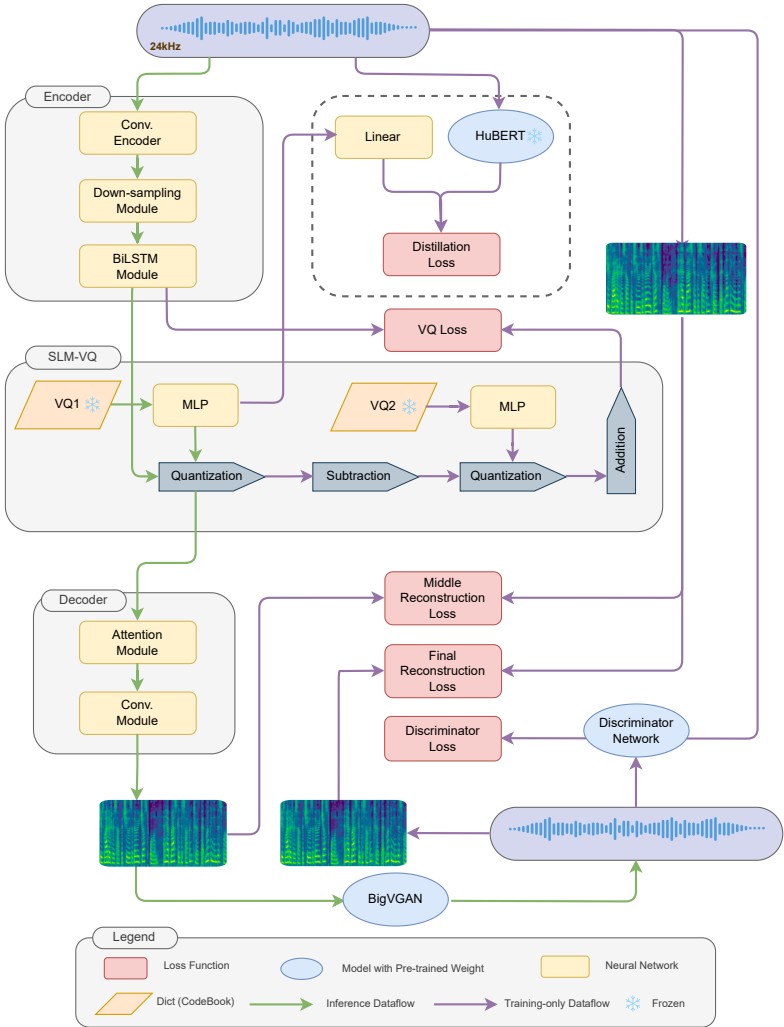

*Figure 1.* The architecture overview of *HH-Codec*. Different color lines indicate the data flow used in inference and only for training. During inference, the audio is processed through the encoder and VQ1 to generate discrete quantization, which is then refined by the MLP. The decoder and fine-tuned BigVGAN subsequently reconstruct the Mel-spectrogram and audio.

at inference. During training, the encoded output $e$ of a 24 kHz input audio $z$ is passed through multi-layer RVQ to obtain the quantized output $\hat{e}$. Both quantizers are updated via the VQ loss $\mathcal{L}_{vq}$ (Eq. 1), where $\mathrm{sg}$ denotes the stop-gradient operator and $\beta$ is a hyperparameter set to 1. However, only the first-layer quantized output is propagated through the decoder for audio reconstruction training and inference. This design preserves the simplicity of single-layer VQ during inference while enhancing representational capacity to accommodate high compression ratios. Conceptually, this approach aligns with techniques such as "Virtual Class" (Chen et al., 2018) and "Register Tokens" (Darcet et al., 2024), where auxiliary learning objectives (such as the second-layer VQ in our case) are introduced to alleviate optimization challenges, thereby enhancing the compact-

ness and efficiency of the first layer. Additionally, instead of using the conventional straight-through estimator (Bengio et al., 2013) for gradient propagation through quantization, we adopt the rotational trick introduced in Fifty et al. (2024). This technique substantively improves reconstruction performance, and increases codebook utilization.

$$\mathcal{L}_{vq} = \|\mathrm{sg}[\mathbf{e}] - \hat{e}\|_2^2 + \beta\|\mathrm{sg}[\hat{e}] - \mathbf{e}\|_2^2. \qquad (1)$$

Since this VQ space is designed to support spoken language models, depending solely on the reconstruction loss for learning explicit semantic and acoustic tokens is challenging. Moreover, the high compression ratio leads to instable training and significant information loss in the early training stages. To address this, we apply semantic model distillation from a pre-trained HuBERT (Hsu et al., 2021) model to the

first quantizer. The distillation objective is formalized in Eq. 2, where $\sigma(\cdot)$ represents the sigmoid function, $D$ denotes the HuBERT feature dimension, $\mathbf{VQ}_1$ represents the output of the first-layer quantizer processed through a linear projection, $\mathbf{H}$ signifies the semantic teacher representations from HuBERT. The function $\cos(\cdot, \cdot)$ computes the cosine similarity between two vectors. By integrating semantic distillation with reconstruction objectives, our approach significantly enhances the process of minimizing information loss, effectively preserving both semantic and acoustic features. Other forms of supervision could be incorporated in a similar framework, though we defer such extensions to future research.

$$\mathcal{L}_{\text{distill}} = -\frac{1}{D} \sum_{d=1}^{D} \log \sigma \left( \cos \left( \mathbf{VQ}_1^{(:,d)}, \mathbf{H}^{(:,d)} \right) \right) \quad (2)$$

### 3.4. Decoder

To ensure consistency in distillation, our encoder takes raw audios as input. However, directly reconstructing audio in a highly compressed codebook presents significant challenges, significantly increasing the difficulty of decoder training. Instead, the Mel-spectrogram provides a more structured representation than raw audio waveforms, as it reduces high-frequency redundancy to enhance low-frequency fidelity while aligning with human auditory perception (Li et al., 2024). Consequently, our entire architecture employs an asymmetric Audio-VQ-Mel-Audio architecture (i.e. more complex decoder). Specifically, we adapt methodologies from Siuzdak (2023) and Ji et al. (2024), incorporating extra attention layers and ConvNeXt blocks to improve Mel-spectrogram reconstruction. Each ConvNeXt block refines input features for reconstruction through stacked 1D convolutions and an inverted bottleneck, with GELU activations and layer normalization further optimizing the decoding process. The reconstructed Mel-spectrogram serves as a structured intermediate output, which is then converted into high-fidelity audio by the BigVGAN (Lee et al., 2022) module. This approach yields superior results compared to alternative encoders using the inverse Fourier transform.

### 3.5. Training Strategies

To optimize the performance upper bound for this complex architecture, we employ dual supervisions using both the Mel-spectrogram and audio representations. Concretely, we denote the middle Mel spectrogram reconstruction as $\text{mel}_{\text{rec}}$, and the final reconstructed audio as $\hat{z}$ with its corresponding Mel spectrogram $\hat{\text{mel}}_{\text{rec}}$. The reconstruction loss $\mathcal{L}_{\text{rec}}$ is composed of three components: the Mel loss $\mathcal{L}_{\text{mel}}$, the adversarial loss $\mathcal{L}_{\text{g}}$, and the feature matching loss $\mathcal{L}_{\text{feat}}$. This approach provides explicit supervision at both critical stages: the decoder-generated (middle) Mel-spectrogram and the (final) Mel-spectrogram obtained from

the BigVGAN-synthesized waveform.

$$\mathcal{L}_{\text{mel}} = \|\text{mel} - \text{mel}_{\text{rec}}\|_1 + \|\text{mel} - \hat{\text{mel}}_{\text{rec}}\|_1 \quad (3)$$

Similar to Kong et al. (2020), the adversarial loss is formulated as the hinge loss of the discriminator logits:

$$\mathcal{L}_{\text{g}} = \frac{1}{K} \sum_{k=1}^{K} \max(1 - D_k(\hat{\mathbf{z}}), 0) \quad (4)$$

Besides, the $\mathcal{L}_{\text{feat}}$ is computed as the mean of the distances between the $l$-th feature maps of the $k$-th distriminator.

$$\mathcal{L}_{feat} = \frac{1}{KL} \sum_{k=1}^{K} \sum_{l=1}^{L} \frac{\|D_k^l(\mathbf{z}) - D_k^l(\hat{\mathbf{z}})\|_1}{mean(\|D_k^l(\mathbf{z})\|_1)} \quad (5)$$

For the discrimination loss, we employ a comprehensive multi-scale discriminator framework combining: (1) a multi-period discriminator (Kong et al., 2020), (2) an STFT discriminator at multiple time scales (Zeghidour et al., 2021), and (3) a Multi-Scale Sub-Band CQT Discriminator (Gu et al., 2024). The discrimination loss $\mathcal{L}_D$ is defined in Eq. 6.

$$\mathcal{L}_D = \frac{1}{K} \sum_{k=1}^{K} \max(1 - D_k(\mathbf{z}), 0) + \max(1 + D_k(\hat{\mathbf{z}}), 0) \quad (6)$$

Directly fine-tuning pre-trained BigVGAN[1] within this paradigm leads to training divergence, as encoder-decoder reconstruction errors in early stages negatively affect the BigVGAN's Mel-to-audio generation process. To mitigate this, we adopt a progressive training strategy consisting of two distinct phases: (1) we initially optimize the middle-stage Mel-spectrogram reconstruction using only $\mathcal{L}_{\text{feat}} + \mathcal{L}_{\text{mel}} + \mathcal{L}_{\text{g}}$, without adversarial training, until the loss value drops below a threshold of 1 (typically after around 20 epochs). (2) We then fine-tune the entire pipeline, with a comprehensive weighted loss function incorporating all previously mentioned objective terms.

$$\mathcal{L} = \lambda_{\text{rec}}(\mathcal{L}_{\text{feat}} + \mathcal{L}_{\text{mel}} + \mathcal{L}_{\text{g}}) + \lambda_{\text{D}}\mathcal{L}_{\text{D}} + \lambda_{\text{distill}}\mathcal{L}_{\text{distill}} + \lambda_{\text{vq}}\mathcal{L}_{\text{vq}} \quad (7)$$

## 4. Experiments

We utilize LibriSpeech train-clean $100/360$ (Zen et al., 2019), VCTK (Veaux et al., 2016), and a subset of the Emilia (He et al., 2024) for our experiments. The Mel spectrogram is computed with a hop length of $256$ and a window length of $1024$. Our *HH-codec* model is trained for $30$ epochs on $4$ A100 GPUs, with a batch size of $6$ and a learning rate of $1 \times 10^{-4}$.

---

[1] https://huggingface.co/nvidia/bigvgan_v2_24khz_100band_256x

*Table 1.* **Reconstruction Results**. $N_q$ denotes the number of quantizers. The origin human voice's UTMOS (Saeki et al., 2022) of three dataset (LibriTTS test-other / LibriTTS test-clean / Seed-TTS-eval) is 3.48 / 4.05 / 3.57.

| Model | Bandwidth ↓ | $N_q$ ↓ | Tokens/s ↓ | UTMOS ↑ | STOI ↑ | V/UV F1 ↑ | SIM ↑ |
|---|---|---|---|---|---|---|---|
| *LibriTTS test-other (noise)* (Zen et al., 2019) | | | | | | | |
| DAC (Kumar et al., 2024) | 9kbps | 9 | 900 | 3.36 | 0.95 | 0.97 | 0.92 |
| SpeechTokenizer (Zhang et al., 2024) | 3kbps | 8 | 600 | 3.28 | 0.87 | 0.92 | 0.79 |
| Encodec (Défossez et al., 2022) | 6kbps | 8 | 600 | 3.02 | 0.91 | 0.93 | 0.68 |
| DAC (Kumar et al., 2024) | 4kbps | 4 | 400 | 2.95 | 0.89 | 0.93 | 0.70 |
| Vocos (Siuzdak, 2023) | 3kbps | 4 | 300 | 3.06 | 0.90 | 0.92 | 0.78 |
| SpeechTokenizer (Zhang et al., 2024) | 3kbps | 4 | 300 | 3.01 | 0.86 | 0.83 | 0.64 |
| Moshi (Défossez et al., 2024) | 1.1kbps | 8 | 104 | 3.06 | 0.85 | 0.88 | 0.68 |
| WavTokenizer-Big Dataset (Ji et al., 2024) | 0.5kbps | 1 | 40 | 3.08 | 0.84 | 0.88 | 0.69 |
| SpeechTokenizer (Zhang et al., 2024) | 0.75kbps | 1 | 75 | 1.27 | 0.73 | 0.60 | 0.38 |
| *HH-Codec* | 0.3kbps | 1 | 24 | 3.21 | 0.86 | 0.86 | 0.71 |
| *LibriTTS test-clean (clean)* (Zen et al., 2019) | | | | | | | |
| DAC (Kumar et al., 2024) | 9kbps | 9 | 900 | 4.03 | 0.97 | 0.97 | 0.94 |
| SpeechTokenizer (Zhang et al., 2024) | 3kbps | 8 | 600 | 3.87 | 0.91 | 0.94 | 0.82 |
| Encodec (Défossez et al., 2022) | 6kbps | 8 | 600 | 3.46 | 0.94 | 0.95 | 0.71 |
| DAC (Kumar et al., 2024) | 4kbps | 4 | 400 | 3.41 | 0.91 | 0.95 | 0.72 |
| Vocos (Siuzdak, 2023) | 3kbps | 4 | 300 | 3.54 | 0.93 | 0.94 | 0.81 |
| SpeechTokenizer (Zhang et al., 2024) | 3kbps | 4 | 300 | 3.49 | 0.87 | 0.86 | 0.69 |
| Moshi (Défossez et al., 2024) | 1.1kbps | 8 | 104 | 3.57 | 0.88 | 0.91 | 0.72 |
| WavTokenizer-Big Dataset (Ji et al., 2024) | 0.5kbps | 1 | 40 | 3.58 | 0.88 | 0.91 | 0.72 |
| SpeechTokenizer (Zhang et al., 2024) | 0.75kbps | 1 | 75 | 1.26 | 0.74 | 0.64 | 0.33 |
| *HH-Codec* | 0.3kbps | 1 | 24 | 3.61 | 0.89 | 0.90 | 0.73 |
| *Seed-TTS-eval (out-of-domain)* (Anastassiou et al., 2024) | | | | | | | |
| DAC (Kumar et al., 2024) | 9kbps | 9 | 900 | 3.46 | 0.96 | 0.99 | 0.91 |
| SpeechTokenizer (Zhang et al., 2024) | 3kbps | 8 | 600 | 3.34 | 0.88 | 0.91 | 0.72 |
| Encodec (Défossez et al., 2022) | 6kbps | 8 | 600 | 2.76 | 0.90 | 0.93 | 0.75 |
| DAC (Kumar et al., 2024) | 4kbps | 4 | 400 | 2.67 | 0.90 | 0.93 | 0.74 |
| Vocos (Siuzdak, 2023) | 3kbps | 4 | 300 | 3.25 | 0.85 | 0.91 | 0.81 |
| SpeechTokenizer (Zhang et al., 2024) | 3kbps | 4 | 300 | 3.11 | 0.88 | 0.87 | 0.70 |
| Moshi (Défossez et al., 2024) | 1.1kbps | 8 | 104 | 3.23 | 0.89 | 0.87 | 0.74 |
| WavTokenizer-Big Dataset (Ji et al., 2024) | 0.5kbps | 1 | 40 | 3.23 | 0.84 | 0.88 | 0.68 |
| SpeechTokenizer (Zhang et al., 2024) | 0.75kbps | 1 | 75 | 1.26 | 0.75 | 0.61 | 0.28 |
| *HH-Codec* | 0.3kbps | 1 | 24 | 3.33 | 0.85 | 0.88 | 0.73 |

## 4.1. Baselines

We compare the reconstruction performance of *HH-Codec* with state-of-the-art codec models, including Vocos (Siuzdak, 2023), Encodec (Défossez et al., 2022), SpeechTokenizer (Zhang et al., 2024), DAC (Kumar et al., 2024), Moshi, and WavTokenizer (Ji et al., 2024). Additionally, for SpeechTokenizer (Zhang et al., 2024) and DAC (Kumar et al., 2024), we evaluated the results using different numbers of quantizers. The reported scores are derived from the official open-source weights provided by each work.

## 4.2. Evaluation Metrics

For evaluation metrics, we employ UTMOS (Saeki et al., 2022) alongside speech-enhancement measures such as Short-time Objective Intelligibility (STOI) (Taal et al., 2010) and the F1 score for voiced/unvoiced classification (V/UV F1). Additionally, we utilize WavLM-large fine-tuned for speaker verification (Anastassiou et al., 2024) to extract

speaker embeddings and compute the cosine similarity (SIM) between reconstructed and original audio.

## 4.3. Main Results

We evaluate our method on three datasets—LibriTTS test-other (Zen et al., 2019), LibriTTS test-clean (Zen et al., 2019), and Seed-TTS-eval (Anastassiou et al., 2024)—to demonstrate its performance under in-domain noisy conditions, in-domain clean conditions, and out-of-domain scenarios. Table 1 shows that, across noisy, clean, and out-of-domain test sets, our codec operating at just 0.3 kbps not only outperforms a model using ten times the bandwidth in UTMOS (Saeki et al., 2022) but also matches its STOI, V/UV F1, and SIM scores. Moreover, when compared with other codecs at similar bitrates, our method exhibits a clear and decisive performance advantage.

| Model | UTMOS ↑ | V/UV F1 ↑ | SIM ↑ |
|---|---|---|---|
| *Trained on a subset of LibriTTS, tested on LibriTTS test-other* | | | |
| *HH-Codec* | 3.07 | 0.87 | 0.64 |
| w/ Classic VQ | 2.76↓ 0.31 | 0.81↓ 0.06 | 0.61↓ 0.03 |
| w/ Single SLM-VQ | 2.94↓ 0.13 | 0.83↓ 0.04 | 0.62↓ 0.02 |
| w/ Fourier decoder | 2.84↓ 0.23 | 0.85↓ 0.02 | 0.62↓ 0.02 |
| w/ Single supervision | 1.85↓ 1.22 | 0.74↓ 0.13 | 0.33↓ 0.31 |
| w/o Progressive Training | 1.88↓ 1.19 | 0.72↓ 0.15 | 0.32↓ 0.32 |
| w/ Simple Network | 2.99↓ 0.08 | 0.85↓ 0.02 | 0.62↓ 0.02 |
| w/o Long windows | 2.94↓ 0.13 | 0.84↓ 0.03 | 0.62↓ 0.02 |

*Table 2.* Evaluation of different *HH-Codec* variants. These ablations support the effectiveness of designs in *HH-Codec*.

| Codebook Size | 1024 | 2048 | 4096 | 8192 | 16384 |
|---|---|---|---|---|---|
| Classic VQ | 99% | 95% | 90% | 56% | 42% |
| Single SLM-VQ | 99% | 98% | 95% | 92% | 87% |
| SLM-VQ | 99% | 98% | 98% | 98% | 94% |

*Table 3.* Codebook utilization of different quantizers under varying codebook sizes. SLM-VQ works well in all settings.

### 4.4. Ablation Study

Due to limited computational resources, we conduct ablation studies by training *HH-Codec* only on the LibriTTS train-100/360 set. Table 2 reports its reconstruction performance evaluated on the LibriTTS test-other set, measured through three metrics, V/UV F1, and SIM. Our systematic ablation experiments demonstrate the necessity of each component and training strategy in our algorithm. The concrete experimental configurations are as follows:

- **SLM-VQ**. We experiment with *w/ Classic VQ* and *w/ Single SLM-VQ*—the latter using a single VQ layer for both training and inference—and report codebook utilization versus codebook size in Table 3.

- **Asymmetric Architecture & Dual-Supervision & Progressive Training** . Similar to most encoder-decoder architectures, we replace the asymmetric framework with a vocoder-like decoder using inverse Fourier transform, denoted as *w/ ·Fourier decoder*. To assess dual supervision stability, we also experiment with supervising only the final audio, denoted as *w/ Single supervision*. We conduct cold-start experiments without Progressive Training, denoted as *w/o Progressive Training*.

- **Neural Network Design**. We replace the decoder's BiLSTM with an LSTM and substitute the discriminator with a simpler version, denoted *w/ Simple Network*. We set the input time window to one second and remove the encoder's attention module, denoted *w/o Long windows*.

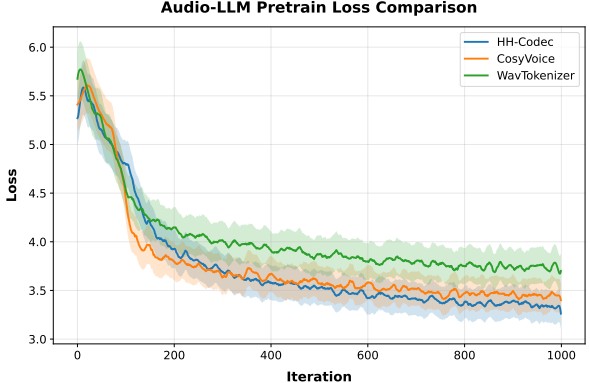

*Figure 2.* Efficient downstream audio-LLM training loss.

### 4.5. Downstream Task: Spoken Language Modeling

To validate the effectiveness of our HH-Codec for spoken language modeling, we conduct ablation studies by integrating this discrete codec with downstream large audio-LLM training. Here we combine *HH-Codec* with Qwen2.5-7B (Team, 2024). Our experiments systematically compare three state-of-the-art audio tokenization approaches: (1) the proposed *HH-Codec*, (2) WavTokenizer (Ji et al., 2024), and (3) CosyVoice (Du et al., 2024), under identical training hyper-parameters and model architectures. As shown in Figure 2, several key findings emerge: at a codebook size of 8192, our method achieves the fastest loss reduction at the lowest cost compared to alternatives, highlighting its effectiveness as a codec for downstream training.

## 5. Conclusion and Discussion

In this work, we present *HH-Codec*, a novel neural codec architecture that achieves extreme speech compression at 24 tokens per second using single-quantizer inference efficiency. By introducing a SLM-VQ space and an asymmetric encoder-decoder architecture, *HH-Codec* delivers state-of-the-art speech reconstruction at a remarkably low bandwidth of 0.3 kbps. Comprehensive ablation studies validate the ef-

fectiveness of each neural network design and training technique. Most significantly, the codec's efficient tokenization scheme and preserved linguistic properties make it particularly suitable for large-scale spoken language model, where it could enable: (1) unified speech-text foundation models through joint embedding spaces, (2) real-time interactive agents with low-latency speech understanding and generation capabilities, and (3) memory-efficient multi-modal systems that maintain conversational context across extended interactions. These directions position *HH-Codec* not just as an audio compression tool, but as a potential enabler for the next generation of interactive speech-enabled AI systems.

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
