# OpenReview forum: "HH-Codec: High Compression High-fidelity Discrete Neural Codec for Spoken Language Modeling"
_ICML.cc/2025/Workshop/TokShop — TokShop_

### Official Review · Reviewer_euiX · 2025-06-07

**Rating:** 7
**Confidence:** 3

**Review:**

This paper aims to train a low-bitrate discrete codec for speech.

Prior works:
1) like Hubert achieve compact representations by sacrificing acoustic information or
2)  use multiple residual quantizers that preserves acoustic information at the expense of higher bitrate.

Before setting out to develop a new method i.e. HH-Codec, they tried to modify existing methods to reduce the token-counts, but observed severe degradation. The authors develop HH-Codec, which achieves a compression rate of 24Hz at a bitrate of 0.3kbps.

Strengths:
- The paper adopts recent advances in learning discrete representations such as the rotation-trick, using longer-training windows than 1 second clips, using multiple RVQ layers but to enhance and stabilize learning, but using only the first layer for reconstruction to keep the bitrate low.
- They use semantic distillation from HuBERT to improve semantic content in the discrete representations.
- They use dual supervision from the audio and its Mel-spectrogram.
- Baselines are exhaustive and recent, including Encodec, Moshi and DAC.
- The method performs very well in terms of UTMOS score, given its low bitrate, almost at par with the other high bitrate methods. The ablations show that all the design choices are essential to performance. The model also achieves very high codebook utilization.
- The method also yields lower loss for AudioLM training compared to other tokenizers (see my comment below about this).

Overall, the method achieves what it has set out to do, i.e. high information retention with a low bitrate.

Weaknesses

- The evaluation metrics reported in this paper are a bit non-standard. The paper reports UTMOS scores, and not ViSQOL or MUSHRA or ABX scores which is typically reported by other papers such as Encodec, Moshi and DAC. Can you authors either comment on why the other scores are not needed, or provide the other scores as well?
- The paper seems to be missing evaluations for semantic knowledge in the tokens.
- Could you authors give some more information on Figure 2 i.e. what steps have been taken to make the loss across the codecs comparable? How can different tokenizations be compared on the same loss curve? Do all methods use a codebook of the same size and number of quantizers?

---

### Official Review · Reviewer_rMq6 · 2025-06-08
**Author(s) suggest a powerful a novel neural codec that achieves extremely high speech compression. This powerful compression codec can help unifying speech and text representations for LMs. The core contribution of achieving 24 token/s with single quantizer can be a great stepping stone making audio more manageable for language models.**

**Rating:** 8
**Confidence:** 5

**Review:**

The author(s) have proposed a powerful neural codec architecture which derives from multiple works and includes novel training strategy.

Key highlights of the work according to me.
1) Achieves Extreme Compression with a Single Quantizer
According to me this is the key contribution of the work. HH-Codec claims to compress high-fidelity audio down to an extremely low rate of 24 tokens per second at just 0.3 kbps. This is phenomenal considering it only uses 1 quantizer during inference. As pointed by author(s) correctly various high quality neural codecs achieve similar(or slightly better) performance but they use multiple quantizers(which make training them very unstable) and they operate at much higher rates (~300-900 tokens/s), which makes them difficult for downstream LM tasks.

2) Introduces Novel Architecture and Training Strategy.
This is a very interesting read and in my opinion that most interesting part of the work. To achieve such compression during test time, the author(s) introduce a sophisticated design which derives multiple neural code, VQ and training strategy works. It is not a single invention but a clever combination multiple good ideas.

Some highlights from my read.
2.1) Introduces an asymmetric architecture encoder / decoder. It introduces a novel workflow Audio -> VQ -> Mel -> Audio.
2.2) A specialized SLM-VQ. This is an interesting and novel idea. The usage of "virtual class" is a good addition.
2.3) Semantic distillation wrt to HuBERT model seem to add linguistic knowledge in tokens.

 The six loss head training strategy, the progressive training and dual supervision ensuring both reconstruction of both Mel and Audio are novel additions.

3) Good experimental results to back the architecture.
The experimental results shows the HHCodec achieve SOTA in reconstruction quality and bitrate. The ablation studies are quite interesting albeit I would have loved to see it done extensively for more datasets. The integration with downstream spoken language modeling task seem to be incomplete without concrete performance metric on a downstream task.

In my opinion the strengths of the paper.
1) The paper's main achievement is 24 token/s with a single quantizer during inference bringing down audio token much closer to text for LMs.
2) The methodology used for designing the system is rigorous. The author(s) identify failure modes like training collapse, low code utilization and propose solution for each.
3) Integration with powerful existing architecture to build an even better(like HuBERT, BigVGAN) architecture is a smart idea.

Some areas of improvement according to me.
1) As per my understanding the computation cost of such system will be higher than the existing neural codec systems(For example due to using BigVGAN). I would have loved to see efficiency metrics like FLOPs for a better tradeoff assessment.
2) Downstream evaluation of LM leaves some scope of improvement. Showing a faster loss reduction is a good start but I would like to see more evaluation of concrete tasks.
3) Paper talks about preservation of audio nuances(for example emotions). I would like to see some evaluation for that.
4) The architecture has complex training procedure involved and its stability has not be fully explored. The ablation studies clearly point to the fact that those additions are necessary and helpful. However such a complex technique will hard to reproduce. Progressive training training dynamics will require careful monitoring of the system. For example, showing the training and validation loss curves across epochs compared to a simpler, single-stage baseline would provide valuable insight for the community.

---

### Decision · Program_Chairs · 2025-06-10

Accept